# The Impact of Electrical Stimulation of the Brain and Spinal Cord on Iron and Calcium-Phosphate Metabolism

**DOI:** 10.3390/brainsci11020156

**Published:** 2021-01-25

**Authors:** Paweł Sokal, Milena Świtońska, Sara Kierońska, Marcin Rudaś, Marek Harat

**Affiliations:** 1Department of Neurosurgery and Neurology, Jan Biziel University Hospital No 2, Ujejskiego 75 Street, 85-168 Bydgoszcz, Poland; m.switonska@cm.umk.pl (M.Ś.); sara.kieronska@biziel.pl (S.K.); marcin.rudas@biziel.pl (M.R.); 2Faculty of Health Science, Collegium Medicum in Bydgoszcz, Nicolaus Copernicus University in Toruń, Jagielonska 13–15 Street, 85-067 Bydgoszcz, Poland; marek.harat@10wsk.mil.pl; 3Department of Neurosurgery, The 10th Military Research Hospital, Powstanców Warszawy 5 Street, 85-081 Bydgoszcz, Poland

**Keywords:** deep brain stimulation, spinal cord stimulation, iron metabolism, c-reactive protein, electric current, calcium phosphate homeostasis

## Abstract

Background: Deep-brain stimulation (DBS) electrically modulates the subcortical brain regions. Under conditions of monopolar cerebral stimulation, electrical current flows between electrode’s contacts and an implantable pulse generator, placed in the subclavicular area. Spinal cord stimulation (SCS) delivers an electrical current to the spinal cord. Epidural electrical stimulation is associated with the leakage of current, which can cause a generalized reaction. The aim of our study was to investigate whether the electrical stimulation of the cerebrum and spinal cord could have generalized effects on biochemical parameters. Materials and methods: A total of 25 patients with Parkinson’s disease (PD, *n* = 21) and dystonia (*n* = 4), who underwent DBS implantation, and 12 patients with chronic pain, who had SCS, received electrical stimulation. The blood levels of selected biochemical parameters were measured before and after overnight stimulation. Results: After DBS, the mean ± interquartile range (IQR) values for iron (off 15.6 ± 13.53 µmol/L; on: 7.65 ± 10.8 µmol/L; *p* < 0.001), transferrin (off: 2.42 ± 0.88 g/L; on: 1.99 ± 0.59 g/L; *p* < 0.001), transferrin saturation (off: 23.20 ± 14.50%; on: 10.70 ± 11.35%; *p* = 0.001), phosphate (off: 1.04 ± 0.2 mmol/L; on: 0.83 ± 0.2 mmol/L; *p* = 0.007), and total calcium (off: 2.39 ± 0.29 mmol/L; on: 2.27 ± 0.19 mmol/L; *p* = 0.016) were significantly reduced, whereas ferritin (off: 112.00 ± 89.00 ng/mL; on: 150.00 ± 89.00 ng/mL; *p* = 0.003) and C-reactive protein (off: 0.90 ± 19.39 mg/L; on: 60.35 ± 35.91 mg/L; *p* = 0.002) were significantly increased. Among patients with SCS, significant differences were observed for ferritin (off: 35 ± 63 ng/mL; on: 56 ± 62 ng/mL; *p* = 0.013), transferrin (off: 2.70 ± 0.74 g/L; on: 2.49 ± 0.69 g/L; *p* = 0.048), and C-reactive protein (off: 31.00 ± 36.40 mg/L; on: 36.60 ± 62.030 mg/L; *p* = 0.018) before and after electrical stimulation. No significant changes in the examined parameters were observed among patients after thalamotomy and pallidotomy. Conclusions: Leaking electric current delivered to the subcortical nuclei of the brain and the dorsal column of the spinal cord exposes the rest of the body to a negative charge. The generalized reaction is associated with an inflammatory response and altered iron and calcium-phosphate metabolism. Alterations in iron metabolism due to electrical stimulation may impact the course of PD. Future research should investigate the influence of electric current and electromagnetic field induced by neurostimulators on human metabolism.

## 1. Introduction

Movement disorders such as Parkinson’s disease (PD), essential tremor (ET) and dystonia can be effectively treated by deep brain stimulation (DBS). DBS in PD has been shown to improve patients’ symptoms, functionality, and activities of daily living [1,2,3,4]. DBS delivers a constant, low, electrical current to a certain, small area of the brain through implanted electrodes connected to an implantable pulse generator (IPG), using a monopolar or bipolar mode of high-frequency stimulation [4]. The IPG is implanted in the subclavian area, in a subcutaneous pocket, and current flows between the IPG and an electrode implanted in the subthalamic nucleus or globus pallidus. DBS is also known to be responsible for local neurochemical effects associated with the release of neurotransmitters, especially dopamine, in the stimulated network [5].

Spinal cord stimulation (SCS) is an approved treatment option for chronic pain [6]. The indications for SCS are neuropathic and ischemic pain syndromes that are refractory to pharmacological therapy [7]. SCS is used to treat symmetrical or asymmetrical pain localized in the trunk and the extremities [8]. SCS delivers an electric current to the spinal cord, and the distribution of the electric current across the dorsal columns induces electrophysiological and regional biochemical changes [9,10,11], including in the cerebrospinal fluid (CSF) [12]. However, epidural electrical stimulation has been associated with the leakage of current, which can affect the whole body. Stimulation using a low- or high-frequency alternating current, with an amplitude not higher than 10 mA, can cause a generalized reaction, which could be inflammatory in nature [13]. To date, few reports have examined alterations in biochemical parameters due to SCS and DBS. Some studies have reported changes in inflammatory and immunological protein expression associated with tonic and burst spinal stimulation, including elevated levels of epidermal growth factor receptor (EGFR) and interleukin-10 (IL-10) levels in the blood after three months of SCS [12,14]. In reports on the impacts of earthing (grounding), which changes the electrical potential of human body significant alterations in iron and calcium-phosphate homeostasis, were demonstrated [15,16]. Consequently, we suspected that comparable alterations in biochemical parameters could be observed after human exposure to electrical stimulation even with a small intensity.

The aim of the present study was to investigate whether electrical stimulation could result in a generalized effect on biochemical parameters, especially iron metabolism and calcium-phosphate homeostasis, which appear to be susceptible to electrical stimulation.

## 2. Materials and Methods

The study protocol was reviewed and approved by the local Bioethics Committee of 138/16 on 5 February 2016. A total of 25 patients (14 men; age: 42–73 years, median: 61 years), who underwent DBS implantation in the subthalamic nucleus (STN, *n* = 21) or internal globus pallidus (GPi, *n* = 4) for the treatment of PD (*n* = 21) or dystonia (*n* = 4), and 12 patients with chronic pain (8 men; age: 34–71 years, median: 55 years), who received SCS, were recruited into the study. Each patient was informed of the study details and written informed consent was obtained from all participants before enrolment in the study. This study was conducted according to the guidelines established by the Helsinki Declaration. The enrolled patients were treated in the Department of Neurosurgery of the 10th Military Research Hospital in Bydgoszcz, Poland, and in the Department of Neurosurgery and Neurology of the University Hospital No. 2 in Bydgoszcz, Poland. After arrival at the hospital, a blood sample was collected from each patient to assess baseline parameters. Patients with PD and dystonia underwent the stereotactic, unilateral, or bilateral implantation of DBS electrodes in the STN or GPi (9 bilateral, 16 unilateral), under local anaesthesia. The subcutaneous placement of the IPG was performed under general anaesthesia. Patients with chronic pain in the lower back and lower limbs underwent one of the following procedures: the percutaneous implantation of one linear lead, with 8 electrodes, on levels Th7–Th10, under local anaesthesia (*n* = 7), with external trial stimulation: the surgical implantation, through a central flavectomy, of a paddle-type electrode (*n* = 4) and an IPG, under general anaesthesia; or the placement of an IPG, under local anaesthesia (*n* = 1). The indications for SCS in the examined group were failed back surgery syndrome (FBSS, *n* = 6), complex regional pain syndrome (CRPS, *n* = 3), and neurogenic pain (*n* = 3). Patients received a non-rechargeable IPG (Precision Novi^TM^-Boston Scientific, Marlborough, MA, U.S.; Proclaim-Abbot, Chicago, IL, USA), a rechargeable IPG (Montage^TM^-Boston Scientific, Marlborough, MA, USA; Prodigy-Abbot, Chicago, USA), or a Freedom 8 (Stimwave, Scottsdale, AZ, USA). After the cranial or spinal surgeries were performed, stimulation was turned on. Tonic stimulation in SCS was adjusted to a level of comfortable paresthesia, with a frequency ranging from 40 to 70 Hz, a pulse width ranging from 100 to 800 ms, and an amplitude ranging from 0.5 to 3.0 mA. The DBS parameters were a frequency of 130 Hz, a pulse width ranging from 60 to 150 µs, and an amplitude from 0.5 to 3.0 mA. The control group consisted of 7 patients (6 men; age: 34–66 years, median: 55 years) who underwent unilateral thalamotomy (*n* = 3) or pallidotomy (*n* = 4) due to symptoms of PD (*n* = 2), dystonia (*n* = 3), or essential tremor (*n* = 2). Blood samples were collected again the day after the implantation and activation of electrical stimulation. Blood samples from patients were drawn in the morning (at both baseline and the day after the activation of SCS/DBS or the day after lesion surgery), under fasted conditions. Venous blood was collected using serum tubes. Serum was obtained by centrifugation. The blood serum concentrations of iron (µmol/L), ferritin (ng/L), transferrin (g/L), calcium (mmol/L), phosphate (inorganic phosphorus, mmol/L), alkaline phosphatase (ALP, mg/L), creatine kinase (CPK, mg/L), and c-reactive protein (CRP, mg/L) were assessed.

### 2.1. Ferritin

The ferritin levels in patient samples were assessed by the University Hospital Laboratory, using Cobas e601 pack. Patient samples were divided into 5 mL tubes using a mechanical separator and Barricor (BD, Becton Dickinson) to prevent artefacts associated with gel separator tubes. After centrifugation at 4000 RCF (g) for 3 min, plasma samples were measured using a Cobas e601 (Roche Diagnostics, Mannheim, Germany) hormone module. Samples were combined with an anti-ferritin monoclonal antibody conjugated to biotin and a second anti-ferritin monoclonal antibody conjugated with a ruthenium complex, to create a sandwich complex. The addition of streptavidin-covered microparticles interacted with biotin, linking the complex to a solid phase. The reaction mixture was then transferred to a measurement cell, and the micro-particles were magnetically attracted to the electrode surface. Voltage applied to the electrode causes chemiluminescence emission, which was measured by a photon counter (photomultiplier).

### 2.2. Transferrin

The transferrin levels in patient samples were assessed by the University Hospital Laboratory. The study was conducted in a Siemens pack (Siemens Healthcare Diagnostics). Samples containing transferrin were incubated with a buffer, and a sample blank determination was performed prior to the addition of the transferrin antibody. In the presence of an excess of an appropriate antibody, the transferrin concentration was measured as a function of turbidity.

### 2.3. C-Reactive Protein (CRP)

The assessment of CRP levels in patient samples was performed by the University Hospital Laboratory. The study was conducted in a Cobas c 501/502 pack. The CRP test was based on a particle-enhanced immunoturbidimetric assay. Human CRP agglutinates were combined with latex particles coated with monoclonal anti-CRP antibodies. The aggregates were determined turbidimetrically. The reference test had a measuring range of 0.1–320 mg/L, and a lower limit of detection of 0.05 mg/L; the CV for within-run precision ranged from 0.62–0.93% across the control samples. The concentration of CRP was determined by the photometric measurement using wavelengths of 525 and 625 nm.

### 2.4. Alkaline Phosphatase (ALP)

The ALP levels in patients samples were measured by the University Hospital Laboratory. The study was conducted in a Cobas c 311 pack (Roche Diagnostics International Ltd., Rotkreuz, Switzerland).

### 2.5. Iron

The iron levels in patient samples were measured by the University Hospital Laboratory. The study was conducted using a Cobas c 501/502 pack (Roche Diagnostics International Ltd., Rotkreuz, Switzerland). The iron profile components were analysed according to standard methods, using a colourimetric assay, according to the manufacturer’s instructions.

### 2.6. Calcium

The study was conducted in Cobas c501/502 pack. Serum was collected using standard sampling tubes, and samples were divided into 5 mL tubes using a mechanical separator and Barricor (BD, Becton Dickinson). The calcium concentration was measured photometrically.

### 2.7. Creatine Kinase

The study was conducted in Cobas c501/502 pack. Serum was collected using standard sampling tubes or tubes containing separating gel. Patient samples were divided into 5 mL tubes using a mechanical separator and Barricor (BD, Becton Dickinson). The standardized method for the determination of CK with activation by NAC was performed.

### 2.8. Statistical Analysis

The IBM SPSS Statistics 25.0 program was used for statistical analysis. To compare biochemical parameter values between two measurements, the Wilcoxon signed-rank test was performed. To parameter values across groups, the Kruskal–Wallis test was used. The level of significance was specified as α = 0.05. The Wilcoxon signed-rank test was used to compare the values of selected biochemical parameters before and after electrical stimulation (DBS and SCS) for the whole sample group.

## 3. Results

In both the DBS and SCS groups, significant differences were observed between the values before and after electrical stimulation for iron, ferritin, transferrin, transferrin saturation (TS), CRP, phosphate, total calcium, and CK. The levels of iron, transferrin, TS, CRP, phosphate, total calcium, and CK were lower after the stimulation than before stimulation, whereas ferritin and CRP were higher after stimulation than before stimulation. Detailed results are presented in Table 1.

A similar analysis was conducted in the control group (in patients before and after thalamotomy or pallidotomy surgery). Elevated levels of CRP were observed in the post-surgical measurement in this group. The results are presented in Table 2 and Figure 1, Figure 2, Figure 3, Figure 4, Figure 5 and Figure 6.

### 3.1. Comparison of Selected Biochemical Parameters before and after Stimulation in the DBS Group

Among the DBS patients, the analysis demonstrated significant differences between iron, ferritin, transferrin, transferrin saturation, CRP, phosphate, and total calcium before and after stimulation (Figure 1, Figure 2, Figure 3, Figure 4, Figure 5 and Figure 6). The levels of iron, transferrin, transferrin saturation, phosphate, and total calcium decreased significantly after electrical stimulation relative to baseline values. After stimulation, ferritin and CRP levels increased (Table 3).

### 3.2. Comparison of Selected Biochemical Parameters before and after SCS

The analysis conducted among patients with SCS demonstrated significant differences between ferritin, transferrin, and CRP levels before and after electrical stimulation. Increased ferritin and CRP levels were observed after stimulation compared with before stimulation. Non-significant differences were observed in calcium phosphate homeostasis parameters between before and after stimulation, as well. The results are presented in Table 4 and Figure 7.

### 3.3. Biochemical Effects of DBS and SCS Compared with the Effects of Lesional Surgery

The Kruskal–Wallis test identified differences in ferritin among the electrical stimulation groups and the lesional surgery group. A post hoc comparison was performed using Dunn’s test, with Bonferroni correction, which revealed that ferritin in the SCS group was lower than that in the DBS group (*p* = 0.003) and that in the control group (*p* = 0.025).

### 3.4. Differences between Levels of Selected Biochemical Parameters after Stimulation Compared with the Lesion Surgery Group

Post hoc analysis revealed that patients with active stimulation, DBS on (*p* = 0.027) and SCS on (*p* = 0.010), had lower ferritin levels than patients after lesional surgery. Patients with DBS had significantly lower levels of transferrin in comparison to patients with SCS (*p* = 0.020). Patients with DBS had higher CRP levels than patients after lesional surgery (*p* = 0.003). These results are shown in detail in Table 5.

## 4. Discussion

The presented study demonstrated elevated levels of ferritin after the electrical stimulation of either the brain or spinal cord, accompanied by reduced transferrin saturation and a concomitant fall in the transferrin levels. In contrast, lesion stereotactic surgery was not associated with significant changes in iron, transferrin, or ferritin levels. These results suggested that DBS and SCS, which are associated with the delivery of an electrical current, not only act locally on the brain and spinal cord but also have a general impact on the whole body and may cause a permanent reaction that activates acute-phase proteins. The resistance of nervous tissue is lower than that in other tissues, making neural tissue a preferential pathway for electric currents. The resistance of the salty tissues beneath the skin is approximately 300 Ohms [13]. Alternating current can spread through the CSF and blood vessels, generating general biochemical changes. For example, electrical currents can cause burns and change blood count parameters and electrolyte levels [17]. SCS devices for which electrodes are placed epidurally in the spinal canal can selectively deliver current to the thoracic spinal cord. The aim is to modulate the targeted fibres of the dorsal column to inhibit the transmission of painful stimuli. However, because the dura has high resistance, any leaking current flows into the CSF, which has a higher conductivity than both the white and grey matter of the spinal cord [18]. The leak and spread of current into the CSF could impact the extracellular environment of the whole body, affecting the biochemical processes conditioned by the membrane potentials of the cells. The intradural placement of electrodes prevents the loss of current to the CSF, dramatically increasing the stimulation efficiency and reducing the power required to stimulate the dorsal columns by more than 90% [19].

Electrical stimulation is a method for the delivery of a negative charge to the body, which alternates bioelectrical processes in the central nervous system and the whole body. One of these processes is the metabolism of iron, which depends on bioelectrical conditions. Ferritin is a positive acute-phase protein which increases in concentration in response to inflammation. In contrast, transferrin is a negative acute-phase protein, the concentration of which decreases in response to inflammation. Changes in the ferritin, iron, and transferrin levels can be attributed to exposure to an electrical current which affects erythrocytes and all organs, including the liver and the muscles. The application of an electrical charge causes changes in the membrane potential, the distribution of ions, and can affect the rate of enzymatic reactions [20]. Evidence suggests that iron metabolism depends on the influence of an electromagnetic field [21]. Changes in iron metabolism may be caused by injury or oxidative stress and are determinants of neurodegenerative disorders [22]. Kwiatek-Majkusiak et al. observed higher serum levels of pro-hepcidin in PD patients treated with DBS compared with control patients treated with pharmacotherapy only. Hepcidin is an acute phase inflammatory response protein that is also involved in iron metabolism, leading to iron retention in macrophages. Hepcidin is primarily synthesized by hepatocytes, which may indicate that DBS, which delivers a high-frequency electrical current to the subcortical nuclei of the brain, is also responsible for a systemic reaction, affecting hepatocytes [23]. Increased blood ferritin concentrations, associated with a concomitant decrease in labile iron and transferrin levels after electrical stimulation, could suggest that DBS results in the relocation of labile iron in safe ferritin stores outside of the brain, reducing the excessive accumulation of iron in the subcortical nuclei. These findings suggest that electrical stimulation may play a protective role, increasing the chelation of free iron by peripheral ferritin stores and reducing iron transportation into the brain during stimulation. Iron metabolism plays an important role in normal brain function, and numerous studies have documented the involvement of iron and altered metabolism in the pathophysiology of neurodegenerative diseases, including PD [24,25,26]. Changes in iron homeostasis during the course of neurodegenerative diseases result in the altered distribution of iron. Iron can be safely stored in lysosomes within soluble and bioavailable ferritin complexes, and labile iron can be incorporated into mitochondrial proteins for energy production [24]. With age, iron levels in the brain, particularly in the basal ganglia, increase due to changes in iron homeostasis [27]. Elevated total iron concentrations in the substantia nigra have been reported in patients with PD [28], and this accumulation of iron in the dopaminergic neurons of substantia nigra may accelerate neurodegenerative processes due to oxidative stress and the production of reactive oxygen species [24]. Excess, unchelated, free iron can lead to neurotoxicity. Even small amounts of labile, divalent iron can initiate reactions that produce free radicals [29], resulting in increased oxidative stress, which has been shown to participate in the neuronal cell death pathway in PD [29,30]. This effect comes from the translocation of transferrin from the peripheral system to the brain. Medeiros et al. observed significantly reduced iron levels in the peripheral blood of PD patients compared with controls, although no significant changes in ferritin and transferrin concentrations were reported. Increased levels of peripheral iron may participate in PD pathogenesis [30]. To minimize the harmful effects of iron overload in the brain, some novel therapeutic approaches have attempted to apply peripheral iron chelators to PD patients [31].

We observed elevated levels of inflammatory markers, reflected by increased CRP levels in response to electrical stimulation applied to the brain and spinal cord of patients with PD, dystonia, and chronic pain. CRP is a recognized marker of inflammation and infection [32]. CRP is an acute-phase protein that rises in response to inflammation and is synthesized primarily in liver hepatocytes, in addition to other cell types, such as macrophages, lymphocytes, adipocytes, smooth muscle cells, and endothelial cells [32]. Increased CRP levels may reflect local inflammatory processes at the stimulation site. Furthermore, elevated CRP levels can be the consequence of a general reaction induced by the electrical current. An active electrical current is associated with the supply of electrons. In an aqueous environment, such as the CSF, mobile negative charges are transferred by OH− groups. Low-frequency electrical stimulation has been shown to induce changes in blood biochemical parameters, such as blood glucose and total cholesterol, in patients with diabetic neuropathy [33]. SCS has also been shown to elevate inflammatory cytokines in the spinal cord [14]. The application of an alternating current by constant current stimulators to combat local resistance can cause minor tissue injury in the spinal cord or brain, resulting in an inflammatory reaction, which can be detected by the upregulation of inflammatory factors, such as CRP, or changes in ferritin and transferrin levels. However, surgery is also an important pro-inflammatory factor, resulting in increased ferritin and CRP levels, and reduced levels of serum transferrin were observed in a study examining the haematological variables following minor and major orthopaedic surgery [34]. To reduce the influence of surgical effects, we evaluated the changes in biochemical parameters after a minor surgery. The percutaneous implantation of a lead electrode and the implantation of DBS electrodes after burr hole trepanation and the implantation of the IPG in a subcutaneous pocket can be qualified as minor surgery. Moreover, the control group, which consisted of patients who underwent stereotactic lesional surgery associated with a burr hole, did not demonstrate significant changes in CRP levels. In addition, creatine kinase, which is an indicator of muscle injury, was not elevated following these procedures, indicating that the application of an electrical current is an additional pro-inflammatory factor, in addition to surgical intervention, which appeared to have a meaningful impact on inflammatory factors, such as CRP and ferritin.

In the DBS group, we observed an increase in total calcium levels and a decrease in phosphate levels, which were significant despite the lack of changes in alkaline phosphatase activity. The changes in total calcium and phosphate were significant in the DBS group and approached significance (*p* = 0.05) in the SCS group. This result suggests that bones might be sensitive to alternations in electric potential [35]. Pulsed electrical fields have been shown to stimulate osteogenic activity, enhancing the bone mineralization process [35].

### Limitations

The primary limitation of this study was the impact of the surgical procedure, which likely had a meaningful influence on the pro-inflammatory factors assessed the day after surgery. The short timeframe in which changes due to electrical stimulation were examined, the small number of participants, and the lack of any phase of off-stimulation should be considered when analysing the results of this study. The study protocol approved by the Ethics Committee allowed for only two blood withdrawals. However, in the small control group, which consisted of patients who underwent a comparable surgical procedure without electrical stimulation, the observed changes alterations were not significant. It is not clear whether increased inflammatory response (expressed in elevated CRP and a positive acute phase protein as ferritin) is connected with the presence of foreign body or it is caused by an electric current alone. Further studies remain necessary to elucidate the generalized effects that electrical current may have on various metabolic pathways, especially among patients who undergo long-term DBS and SCS. However, changes in biochemical parameters may not be observed in long-term treatment patients due to habituation to electrical stimulation and the development of an altered homeostatic balance.

## 5. Conclusions

The exposure of the human body to a leaking electric current, delivered by DBS or SCS, may cause a generalized response, which is associated primarily with an inflammatory reaction, altered iron metabolism, and changes in calcium phosphate homeostasis. A leaking electric current delivered to either the subcortical nuclei of the brain or the dorsal column of the spinal cord exposes the body to a negative charge. Changes in iron metabolism caused by electrical stimulation may impact the course of PD. Future research should investigate the influence of electric currents and electromagnetic fields induced by neurostimulators on electrolyte homeostasis and delayed metabolic and molecular effects.

## Figures and Tables

**Figure 1 brainsci-11-00156-f001:**
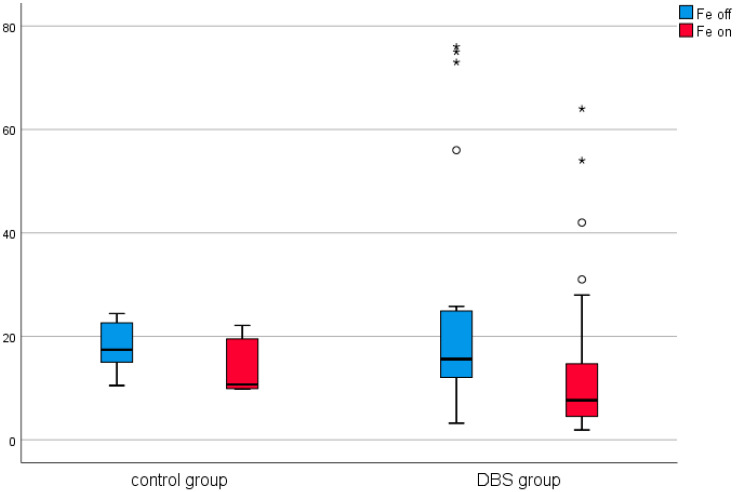
Effect of lesional surgery and deep brain stimulation (DBS) on changes in iron levels (µmol/L). (°—outliers ≦ 3SD, *—outliers > 3SD).

**Figure 2 brainsci-11-00156-f002:**
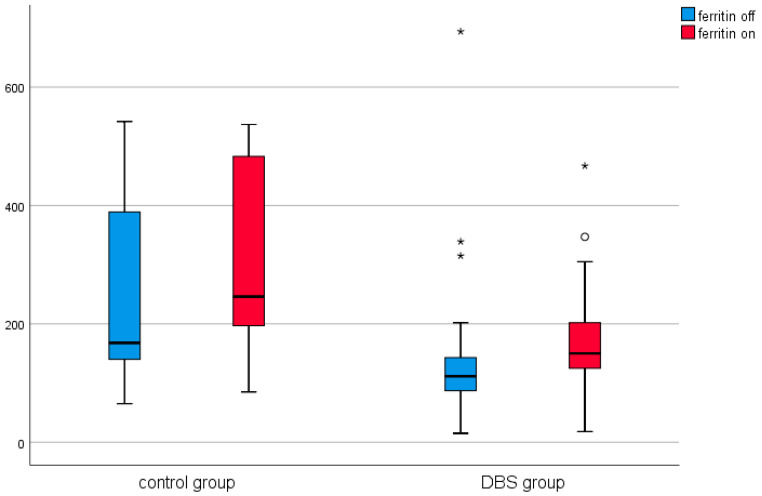
Effect of lesional surgery and DBS on ferritin levels (ng/mL). (°—outliers ≦ 3SD, *—outliers > 3SD).

**Figure 3 brainsci-11-00156-f003:**
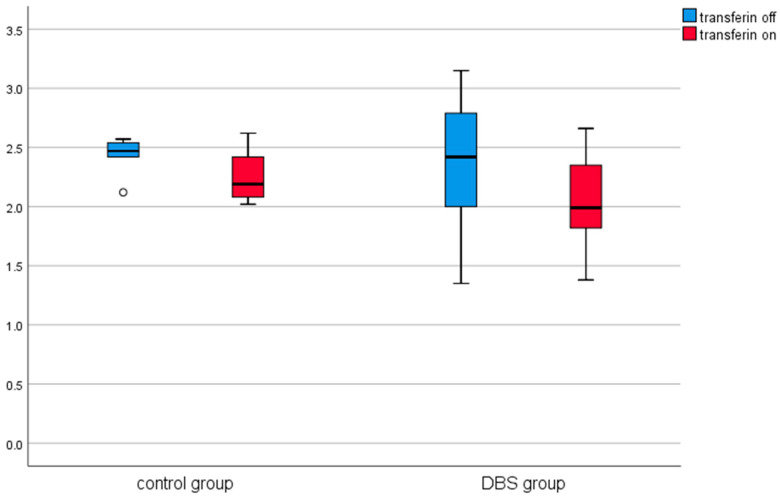
Effect of lesional surgery and DBS on transferrin levels (g/L). (°—outliers ≦ 3SD, *—outliers > 3SD).

**Figure 4 brainsci-11-00156-f004:**
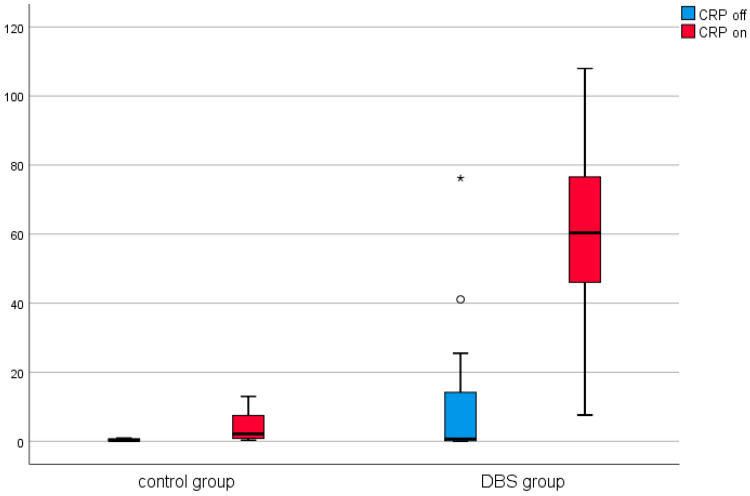
Effect of lesional surgery and DBS on C-reactive protein (CRP) levels (mg/L), (°—outliers ≦ 3SD, *—outliers > 3SD).

**Figure 5 brainsci-11-00156-f005:**
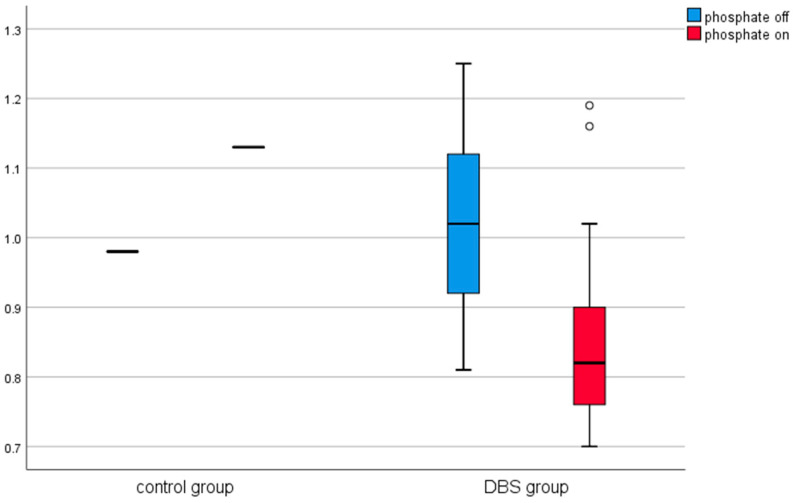
Effect of lesional surgery and DBS on phosphate levels (mmol/L). (°—outliers ≦ 3SD, *—outliers > 3SD).

**Figure 6 brainsci-11-00156-f006:**
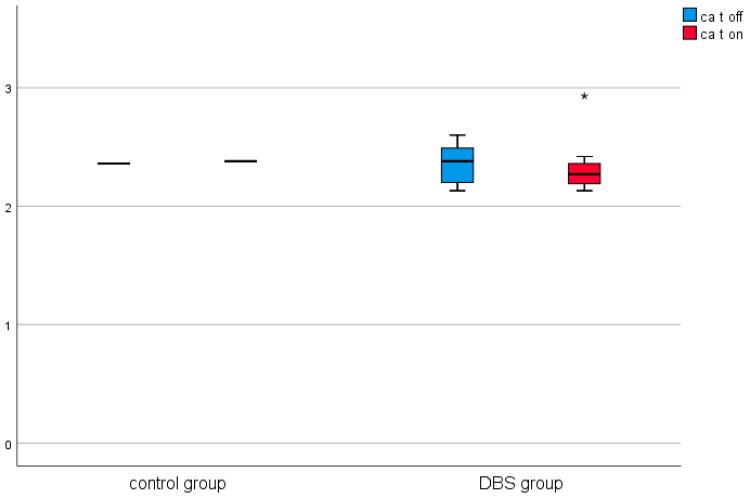
Effect of lesional surgery and DBS on calcium (mmol/L). (°—outliers ≦ 3SD, *—outliers > 3SD).

**Figure 7 brainsci-11-00156-f007:**
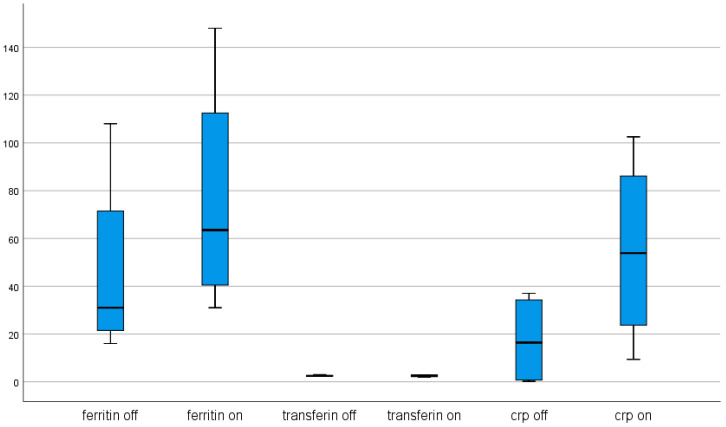
Effect of spinal cord stimulation (SCS) on ferritin (ng/L), transferrin (g/L), and CRP (mg/L) concentrations.

**Table 1 brainsci-11-00156-t001:** Comparison of selected biochemical parameters before and after electrical stimulation in DBS and SCS groups. Parameters in bold were significantly different. Trend arrows show the change of examined parameters.

Parameters		Baseline	Stimulated	*Z*	*p*	*r*
	*Me*	*IQR*	*Me*	*IQR*
**Iron (µmol/L)**	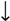	**15.70**	**13.65**	**9.70**	**20.75**	**−3.88**	**<0.001**	**0.43**
**Ferritin (ng/L)**		**92.00**	**104.50**	**130.00**	**135.00**	**−4.49**	**<0.001**	**0.50**
**Transferrin (g/L)**	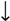	**2.48**	**0.68**	**2.19**	**0.55**	**−4.83**	**<0.001**	**0.51**
**TS%**	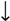	**23.20**	**11.40**	**13.70**	**12.95**	**−2.95**	**0.003**	**0.37**
**CRP (mg/L)**		**0.82**	**26.68**	**38.00**	**61.18**	**−4.46**	**<0.001**	**0.62**
**Phosphate (mmol/L)**	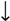	**1.05**	**0.27**	**0.91**	**0.24**	**−2.12**	**0.034**	**0.29**
**Calcium total (mmol/L)**	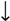	**2.38**	**0.27**	**2.23**	**0.18**	**−3.21**	**0.001**	**0.43**
Calcium ionized (mmol/L)		1.20	0.04	1.17	0.04	−0.86	0.393	0.15
Alkaline phosphatase (u/L)		66.50	13.50	67.00	34.00	−0.47	0.637	0.09
**Creatine kinase (mg/L)**		**112.00**	**394.00**	**91.00**	**350.00**	**−2.03**	**0.043**	**0.54**

DBS: deep brain stimulation; SCS: spinal cord stimulation; Me: mean; IQR: interquartile range; TS: transferrin saturation; CRP: C-reactive protein.

**Table 2 brainsci-11-00156-t002:** Comparison of selected biochemical parameters in the control group (results before and after lesional surgery).

Parameters	Before		After		*Z*	*p*	*r*
	*Me*	*IQR*	*Me*	*IQR*			
Iron (µmol/L)	17.40	10.75	10.70	10.95	−1.75	0.080	0.55
Ferritin (ng/L)	154.00	373.00	246.00	369.00	−1.75	0.080	0.55
Transferrin (g/L)	2.47	0.20	2.19	0.41	−1.57	0.116	0.45
TS (%)	27.10	21.20	17.65	25.18	−1.21	0.225	0.38
CRP (mg/L)	0.28	0.86	2.16	8.19	−2.20	0.028	0.64

Me: mean; IQR: interquartile range; TS%: transferrin saturation; CRP: C-reactive protein.

**Table 3 brainsci-11-00156-t003:** Comparison of selected biochemical parameters before and after stimulation in the DBS group. Trend arrows show the change of examined parameters.

Parameters		Off	On	*Z*	*p*	*r*
	*Me*	*IQR*	*Me*	*IQR*
**Iron (µmol/L)**	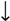	**15.60**	**13.53**	**7.65**	**10.80**	**−4.00**	**<0.001**	**0.58**
**Ferritin (ng/L)**		**112.00**	**89.00**	**150.00**	**89.00**	**−2.94**	**0.003**	**0.44**
**Transferrin (g/L)**	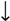	**2.42**	**0.88**	**1.99**	**0.59**	**−3.59**	**<0.001**	**0.51**
**TS (%)**	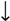	**23.20**	**14.50**	**10.70**	**11.35**	**−3.18**	**0.001**	**0.52**
**CRP (mg/L)**	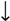	**0.90**	**19.39**	**60.35**	**35.91**	**−3.06**	**0.002**	**0.62**
**Phosphate (mmol/L)**	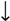	**1.04**	**0.21**	**0.83**	**0.20**	**−2.70**	**0.007**	**0.48**
**Calcium total (mmol/L).**	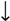	**2.39**	**0.29**	**2.27**	**0.19**	**−2.42**	**0.016**	**0.42**
Calcium ionized (mmol/L).		1.20	0.04	1.18	0.06	−0.67	0.506	0.13
Alkaline phosphatase (u/L)		65.00	28.50	69.00	34.25	−0.59	0.553	0.15
Creatine kinase (mg/L)		107.00	692.75	100.00	599.75	−1.10	0.273	0.45

DBS: deep brain stimulation; Me: mean; IQR: interquartile range; TS: transferrin saturation; CRP: C-reactive protein. Parameters in bold were significantly different.

**Table 4 brainsci-11-00156-t004:** Comparison of selected biochemical parameters before and after SCS.

Parameters		Off	On	*Z*	*p*	*r*
	*Me*	*IQR*	*Me*	*IQR*
Iron (µmol/L)		17.65	43.78	25.50	33.85	−0.46	0.646	0.10
**Ferritin (ng/L)**		**35.00**	**63.00**	**56.00**	**62.00**	**−2.50**	**0.013**	**0.53**
**Transferrin (g/L)**	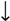	**2.70**	**0.74**	**2.49**	**0.69**	**−2.19**	**0.028**	**0.49**
TS% (%)		17.90	15.80	10.00	31.29	−0.41	0.686	0.13
**CRP (mg/L)**		**31.00**	**36.40**	**36.60**	**62.30**	**−2.37**	**0.018**	**0.63**
Phosphate (mmol/L)		1.05	0.45	1.00	0.17	−0.53	0.594	0.12
Calcium total (mmol/L).		2.35	0.27	2.20	0.13	−1.96	0.050	0.46
Ca ionized (mmol/L).		1.20	0.01	1.17	0.01	−0.45	0.655	0.23
Alkaline phosphatase (u/L)		68.00	6.75	68.00	14.75	−0.18	0.854	0.06
Creatine kinase (mg/L)		112.00	3.00	91.00	16.00	−1.60	0.109	0.65

SCS: spinal cord stimulation; Me: mean; IQR: interquartile range; TS: transferrin saturation; CRP: C-reactive protein. Trend arrows show the change of examined parameters. Parameters in bold were significantly different.

**Table 5 brainsci-11-00156-t005:** Comparison of selected biochemical parameters after electrical stimulation.

Parameters	Lesions		DBS		SCS		*Χ^2^*	*p*	*η^2^*
	*Me*	*IQR*	*Me*	*IQR*	*Me*	*IQR*			
Iron on (µmol/L)	10.70	10.95	7.65	10.80	25.50	33.85	4.45	0.217	0.11
**Ferritin on (ng/L)**	**246.00**	**369.00**	**150.00**	**89.00**	**56.00**	**62.00**	**13.57**	**0.004**	**0.34**
**Transferrin on**	**2.19**	**0.41**	**1.99**	**0.59**	**2.49**	**0.69**	**9.16**	**0.027**	**0.21**
TS on (%)	17.65	25.18	10.70	11.35	10.00	31.29	3.37	0.338	0.10
**CRP on (mg/L)**	**2.16**	**8.19**	**60.35**	**35.91**	**36.60**	**62.30**	**12.09**	**0.007**	**0.47**
Phosphate on (mmol/L)	1.13	0.42	0.83	0.20	1.00	0.17	5.02	0.171	0.17
Ca total on (mmol/L).	2.38	0.00	2.27	0.19	2.20	0.13	3.64	0.303	0.13
Ca ionized on (mmol/L)	-	-	1.18	0.06	1.17	0.01	0.09	0.958	0.01
Alkaline phosphatase on (u/L)	42.00	0.00	69.00	34.25	68.00	14.75	0.77	0.679	0.06
Creatine kinase on (mg/L)	-	-	100.00	599.75	91.00	16.00	0.13	0.724	0.02

DBS: deep brain stimulation; SCS: spinal cord stimulation; Me: mean; IQR: interquartile range. Parameters in bold were significantly different.

## Data Availability

The data presented in this study are available on request from the corresponding author. The data are not publicly available due to data privacy regulations.

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
