# Peer review of "The Impact of Electrical Stimulation of the Brain and Spinal Cord on Iron and Calcium-Phosphate Metabolism"

_brainsci, 2021, doi:10.3390/brainsci11020156_

Round 1

Reviewer 1 Report

The topic of this paper is interesting. There are, however, many drawbacks of the study.

  • the number of patients is too low
  • there is NO control-group (unless one accepts that the patients are there own control due to the longitudinal character)
  • the clinical significance of the findings are not clear at all. Do we make these patients ill by exposing them to this form of stimulation?

Minor:

- pulse-width (at least in DBS) is between 60 and 180us. It is a factor of thousand to ms. 

Author Response

Response for the review nr 1

The topic of this paper is interesting. There are, however, many drawbacks of the study.

  • the number of patients is too low
  • there is NO control-group (unless one accepts that the patients are there own control due to the longitudinal character)

I thank the reviewer for the comments indicating drawbacks of our study.

The number of patients of patients who underwent DBS was 25, the number of patients who underwent SCS was 12. The overall number of patients who underwent surgery and electrical stimulation was 37. This study was designed to visualize alterations in iron and calcium-phosphate metabolism in longitudinal character the day after surgery and electrical stimulation, on the other hand patients who underwent surgery alone without electrical stimulation had no significant changes in these parameters although they constituted the group of 7 people who had surgery similar to DBS surgery. Statistical significance of observed results was demonstrated despite relatively small samples. This was a cohort study in which samples were exposed on DBS and SCS. We find it as a pilot study and observed findings could form the basis for future investigations.

  • the clinical significance of the findings are not clear at all. Do we make these patients ill by exposing them to this form of stimulation?

I agree that the findings need further confirmation and investigation to clarify long-term metabolic effects of electrical stimulation of brain and spinal cord. It seems that DBS and SCS is not neutral for human organism. Not only it helps to reduce clinical symptoms of movement disorders and chronic pain but at the same time can have general influence on human metabolism. These changes are small and not evident due to small currents induced by IPGs but they do exist. Therefore, this report is worth of spreading among clinicians to let them know that they can observe metabolic changes after DBS and SCS. There is no evidence that these changes make these patients ill. Contrary I think that these changes in iron metabolism are beneficial in the course of PD since they could help to reduce iron accumulation in subcortical nuclei but this is only speculation which needs further investigation.

Minor:

- pulse-width (at least in DBS) is between 60 and 180us. It is a factor of thousand to ms. 

It was a mistake, it should be in µs. Thank you. The pulse width applied in clinical practice is different, usually in PD we set 60µs sometimes it increased to 90µs in tremor and further widening is applied in dystonia.

Thank you for valuable input

Reviewer 2 Report

The topic of the paper is very interesting, mainly the idea that electrical current delivered to the brain or to the spinal cord (spinal canal structures) can lead to leakage and generalized biochemical effects.

On the other hand, it is not clear the choice of evaluated parameters and the timing of evaluation. Even if some parameters are significantly changed in the on phase, the great variability render the conclusions uncertain.

In the objective of the study, you should specify the reason (and references) for choosing the biochemical parameters to study. Iron metabolism has been studied (in your references) in relation to surgical conditions and calcium-phosphate in relation electro-magnetic fields applied for osteomalacia/osteoporosis, two areas completely different from neurostimulation.

Even if it is easy to understand the ethical problems, the results would have been more reliable if blood samples were taken one day after surgery with stimulation off and again the second day after surgery with stimulation on in order to differentiate the effect of surgery from the effect of stimulation. The control group received surgery but not an implant and the presence of a foreign body could be the cause of an increased inflammatory response, as demonstrated by CRP increase.

As you state in the conclusion, it would be interesting to re-evaluate the parameters after some months from the implant, but the data would be more reliable with stimulation on and after some time of stimulation off (not feasable for ethical reasons) because of the possible variability of the studied parameters over time.

Author Response

Response for the review nr 2

The topic of the paper is very interesting, mainly the idea that electrical current delivered to the brain or to the spinal cord (spinal canal structures) can lead to leakage and generalized biochemical effects.

On the other hand, it is not clear the choice of evaluated parameters and the timing of evaluation. Even if some parameters are significantly changed in the on phase, the great variability render the conclusions uncertain.

In the objective of the study, you should specify the reason (and references) for choosing the biochemical parameters to study. Iron metabolism has been studied (in your references) in relation to surgical conditions and calcium-phosphate in relation electro-magnetic fields applied for osteomalacia/osteoporosis, two areas completely different from neurostimulation.

As we stated in the article the aim of the present study was to investigate whether electrical stimulation could result in a generalized effect on biochemical parameters, especially iron metabolism and calcium-phosphate homeostasis, which appear to be susceptible to electrical stimulation. The inspiration for these studies was the effect of earthing (grounding) on human physiological parameters as iron metabolism in which slight change in electric potential of the human body (flow of negative charge from the Earth) can dramatically change iron metabolism , electrolytes’ concentrations and calcium-phosphate homeostasis, and even protein levels therefore stimulation with electric current which is connected with the supply of negative charge can be important factor having influence on regulation of metabolism.

In order to clarify the rationale  for choosing the biochemical parameters to study we added the following sentences with two references:  In reports on the impacts of earthing (grounding), which can change the electrical potential of human body significant alterations in iron and calcium-phosphate homeostasis were demonstrated [15][16]. Consequently, we suspected that comparable alterations in biochemical parameters could be observed after exposure of human body to electrical stimulation even with a small intensity.

Sokal, K.; Sokal, P. Earthing the human body influences physiologic processes. J. Altern. Complement. Med. 2011, 17, doi:10.1089/acm.2010.0687.

Chevalier, G.; Sinatra, S.T.; Oschman, J.L.; Sokal, K.; Sokal, P. Earthing: Health implications of reconnecting the human body to the Earth’s surface electrons. J. Environ. Public Health 2012, 2012, doi:10.1155/2012/291541, doi:10.1155/2012/291541.

Even if it is easy to understand the ethical problems, the results would have been more reliable if blood samples were taken one day after surgery with stimulation off and again the second day after surgery with stimulation on in order to differentiate the effect of surgery from the effect of stimulation. The control group received surgery but not an implant and the presence of a foreign body could be the cause of an increased inflammatory response, as demonstrated by CRP increase.

I find this aspect very interesting and fascinating and I am grateful for this comment. We are not sure whether increased inflammatory response (expressed in elevated CRP and positive acute phase protein as ferritin) is connected with the presence of foreign body or it is caused by electric current alone or maybe reaction on foreign body additionally enhanced by an electric current. This phrase has been added to the main text in Limitations section. This aspect needs further studies. It was confirmed that patients with DBS had higher CRP levels than patients after lesional surgery (p = 0.003). Overall, no matter what is the reason of achieved alterations in iron metabolism and calcium-phosphate metabolism it is worth to know that these changes are observed within 24 hours after surgery and at least 12 hours of deep brain stimulation or spinal cord stimulation.

As you state in the conclusion, it would be interesting to re-evaluate the parameters after some months from the implant, but the data would be more reliable with stimulation on and after some time of stimulation off (not feasable for ethical reasons) because of the possible variability of the studied parameters over time.

I totally agree that it would be interesting to re-evaluate these parameters after some months from the implant and we will try to arrange such studies but presented study could be completed in the neurosurgical ward which is dedicated to neurosurgery and patients stay in hospital for a relatively short period of time. It is easier to receive consent from the patient on participation in the study if it is accomplished in a routine hospital stay and is not connected with the prolongation of this stay.

Thank you for important comments 

Round 2

Reviewer 1 Report

Thank you for addressing my concerns.